# Reducing Cd and Pb Accumulation in Potatoes: The Role of Soil Passivators in Contaminated Mining Soils

**DOI:** 10.3390/life14121615

**Published:** 2024-12-06

**Authors:** Lijuan Wang, Hongyin Zhou, Ke Yang, Ladu Er Ze, Zhengli Lu, Yingmei Li, Liyuan Mu, Naiming Zhang

**Affiliations:** 1College of Resources and Environment, Yunnan Agricultural University, Kunming 650201, China; 18387624073@163.com (L.W.); yangke139175@163.com (K.Y.); w18388686246@126.com (L.E.Z.); 15969060608@163.com (Z.L.); 18895890735@163.com (Y.L.); 13038653382@163.com (L.M.); 2College of Plant Protection, Yunnan Agricultural University, Kunming 650201, China; zhy1605202632@163.com

**Keywords:** mining area, passivator, potato, cadmium and lead, combined pollution, accumulation

## Abstract

This work aimed to explore safe techniques for the utilization of farmland surrounding mining areas contaminated with heavy metals—specifically cadmium (Cd) and lead (Pb)—in order to achieve food security in agricultural production. A potato variety (Qingshu 9) with high Cd and Pb accumulation was used as the test crop, and seven treatments were set up: control (CK), special potato fertilizer (T1), humic acid (T2), special potato fertilizer + humic acid (T3), biochar (T4), calcium magnesium phosphate fertilizer (T5), and biochar + calcium magnesium phosphate fertilizer (T6). The remediation effect of the combined application of different passivators on the accumulation of cadmium and lead in potatoes in the contaminated soil of a mining area was studied. The results showed that, compared with CK, all passivator treatments improved the physical and chemical properties of the soil and reduced the available Cd and Pb content in the soil and in different parts of potatoes. The T6 treatment yielded the most significant reduction in the available Cd and Pb content in the soil, the Cd and Pb content in the potato pulp, and the enrichment factor (BCF) and transfer factor (TF) of the potatoes. Compared with T4 and T5, the content of available Cd in the soil decreased by 1.22% and 4.71%, respectively; the soil available Pb content decreased by 3.13% and 3.02%, respectively; the Cd content in the potato pulp decreased by 68.08% and 31.02%, respectively; and the Pb content decreased by 31.03% and 20.00%, respectively. The results showed that the application of biochar combined with calcium magnesium phosphate fertilizer had a better effect in terms of reducing the available Cd and Pb content in the soil and the Cd and Pb content in the potato flesh compared to their individual application. Biochar and calcium magnesium phosphate fertilizer can synergistically increase the content of soil available nutrients and reduce the activity of heavy metals in the soil to prevent the transfer and accumulation of cadmium and lead to potatoes, as well as improve their yield and quality. The results of this study provide technical support for safe potato planting and agricultural soil management.

## 1. Introduction

With the long-term processes of industrialization and urbanization, human activities such as mineral resource extraction, wastewater irrigation, and the improper application of fertilizers and pesticides have led to the introduction of pollutants containing various heavy metal elements into the soil through multiple pathways, resulting in heavy metal contamination [1]. At present, the world is facing the environmental challenge of soil heavy metal pollution, and this problem is widespread in mining areas across Europe [2], Africa [3], and South America [4]. According to the United Nations Food and Agriculture Organization (FAO) and the United Nations Environment Programme (UNEP), approximately 10% of the world’s arable land is affected by heavy metal pollution [5], posing a threat to food security and human health. China is one of the world’s primary mining countries, with a total mining area of approximately 104,000 km^2^, according to statistics from the China Geological Survey [6]. As mineral resources are extensively extracted, significant amounts of mining waste are generated, placing substantial pressure on the environment [7]. Soil heavy metal pollution caused by metal mines is particularly prominent in Southwestern, Eastern, and Southern China [8], with cadmium (Cd) and lead (Pb) as the main pollutants [9,10,11]. Cadmium (Cd) and lead (Pb) affect cell metabolism by destroying plants’ reactive oxygen metabolism systems and antioxidant systems [12,13], thereby inhibiting their growth [14]. At the same time, Cd and Pb enter the human body through the food chain and accumulate, endangering human health [15,16]. Therefore, reducing the absorption and transport of Cd and Pb by crops is an important means to ensure food security.

Potato (*Solanum tuberosum* L.) is the world’s fourth-largest food crop after maize (*Zea mays*), wheat (*Triticum aestivum*), and rice (*Oryza sativa*) [17]. It is characterized by high yields, strong resistance to adversity, and broad adaptability, making it an important raw material for both food and industrial use [18]. Potato has been widely planted and consumed due to its short growth cycle, low cost, and simple planting management technology. However, the root structure of potato is relatively shallow and extensive, which makes it easier to interact with heavy metals in the soil during the crop’s growth, and the degree of toxicity caused by heavy metal pollution is greater than in other crops [19]. Therefore, it is highly important to identify a management strategy to prevent potato from absorbing and accumulating heavy metals in the soil during its growth.

At present, the main methods for the treatment and restoration of farmland polluted by heavy metals are the soil replacement method, electrodynamic remediation method, and bioremediation method. However, these methods have the disadvantages of requiring extensive engineering efforts, carrying high costs, and creating secondary pollution [20]. The passivation method can not only change the speciation of heavy metals in the soil and reduce their mobility and bioavailability; it also has the advantages of economic efficiency, simple operation, and no secondary pollution [21,22]. The materials commonly used in passivation methods include biochar, humic acid, and calcium magnesium phosphate fertilizer. For example, Cheng et al. [23] showed that the application of biochar could effectively reduce the content of heavy metals in plant tissue and increase the plant biomass. Zhang et al. [24] showed that the application of a humic acid water-soluble fertilizer at 600~900 kg hm^−2^ could reduce the content of Cd and Pb in rice. Luo et al. [25] found that the application of a calcium magnesium phosphate fertilizer reduced the bioavailable concentration of cadmium (Cd) in soil by 28.57%.

Most of the studies on the remediation of soil heavy metal pollution using passivators are limited to the use of single-passivator materials [26,27]. In many cases, the remediation effect of a single passivator is poor, particularly when dealing with soil with more complex or serious pollution. The synergistic effects and remediation mechanisms of the combined use of different soil passivators to reduce heavy metal contamination in soil have not been reported. Based on this, the present study selected a special potato fertilizer, humic acid, biochar, and a calcium magnesium phosphate fertilizer as test passivators. It studied the remediation effect of the combined application of different soil passivators on Cd- and Pb-contaminated soil in a mining area and the absorption and accumulation of Cd and Pb in different parts of potato through field experiments. We aimed to determine whether the combined application of different passivators was more effective than the use of a single passivator in preventing the absorption and transfer of heavy metals in potato. Based on this goal, we hypothesized that (1) compared with CK, all passivators would significantly mitigate the Cd and Pb pollution in the soil in the mining area and significantly reduce the Cd and Pb content in various parts of potato; (2) the effect of the combined application of different passivators in terms of reducing the Cd and Pb content in different parts of potato and preventing the absorption and accumulation of Cd and Pb in potato would be stronger than that of a single passivator.

## 2. Materials and Methods

### 2.1. Overview of the Study Area

The study area was located in the typical mining region surrounding the potato cultivation zone of Zhehai Town, Huize County, Qujing City, Yunnan Province, China (26°51′ N, 103°30′ E). Zhehai Town features the deepest lead–zinc mine in Asia and serves as the largest basin in the county, covering an area of 55.6 km^2^. The area is characterized by the karst topography of the Yungui Plateau, with red soil as the predominant soil type. The average elevation is 2099 m, with a mean annual temperature of 12.6 °C and average annual precipitation of 847.1 mm. From a comparison of the national soil pollution risk using the Cd and Pb control values at pH ≤ 5.5 from the GB 15618-2018 [28] “Soil Environmental Quality Agricultural Land Soil Pollution Risk Control Standards (for trial implementation)” (Cd ≤ 1.5 mg·kg^−1^, Pb ≤ 400 mg·kg^−1^), the levels of Cd in the study area were significantly above the recommended levels. Therefore, it is essential to implement safe utilization practices for heavy metal-contaminated soils in this region. The basic physicochemical properties of the soil, along with the concentrations of Cd and Pb, are presented in Table 1.

### 2.2. Test Materials

Test Plant: The potato variety Qing Shu 9 [29], which exhibits high levels of lead and cadmium, was selected from a group of low-accumulation varieties described in a previous study by the research group. This variety was provided by the Nanjing Institute of Soil Science, Chinese Academy of Sciences.

Test Passivators: Four types of passivators were selected for the experiment, namely, a special potato fertilizer, humic acid, a calcium magnesium phosphate fertilizer, and biochar. These were purchased from Yunnan Farmhouse Group Co., Ltd. The special potato fertilizer had a pH value of 5.7 and a N_5_:P_2_O_5_:K_2_O ratio of 16:10:6. As a conventional local fertilizer, it can improve and balance the soil nutrient content. Humic acid’s pH value is 4.5. As a natural colloidal organic substance, humic acid has rich surface functional groups and a dense pore structure, which can promote the adsorption of Cd in soil [30]. The main components of the calcium magnesium phosphate fertilizer were Ca_3_(PO_4_)_2_, CaSiO_3_, and MgSiO_3_, and the pH value was 8.5. Its mechanism of action involved reacting with heavy metal ions to form stable phosphate minerals and reduce the activity of heavy metals in soil [31]. The raw material in the biochar was corn straw, which was produced via pyrolysis at a high temperature (500 °C); the specific surface area was 500 m^2^/g and the pH value was 9.2. The unique large specific surface area and porous structure of biochar mean that it can adsorb heavy metal ions in soil and improve its physical and chemical properties [32].

### 2.3. Experimental Design

The field experiment was conducted from April to August 2023. Healthy and disease-free crops of the local main potato variety “Qingshu No. 9” were selected. They were cut into uniform, small pieces (each piece weighed about 30–50 g), and then these tubers were sown with a row spacing of 80–90 cm and plant spacing of 30–40 cm. The experiment consisted of 7 passivation agent treatments (Table 2); each treatment was repeated three times, and a random block design was used. A total of 21 plots were designed; each plot’s area was 5 m × 6 m = 30 m^2^. Protective rows were established around the experimental plots, and the cultivation management practices during the planting period were consistent with the local practices. All passivator treatments were applied in the form of a base fertilizer before potato planting, and the potato was harvested at the maturity stage at 120 days.

### 2.4. Sample Collection, Measurement, and Analysis

#### 2.4.1. Sample Collection

Potato Sample Collection: Potato plant samples were collected using the five-point sampling method, with five whole plants taken from each plot. After returning to the laboratory, the potato samples were divided into four parts: the roots, stems, leaves, and tubers. The samples were first washed thoroughly with tap water to remove soil and debris, followed by three rinses with distilled water. Once air-dried, the fresh weight was recorded. Each part of the potato was then cut into smaller pieces using a quartz knife and placed into mesh bags. The samples were subjected to a blanching process in an oven at 105 °C for half an hour, after which the temperature was adjusted to 65 °C to dry the samples until a constant weight was achieved, and the dry weight was recorded. The potato parts were then ground using a grinder, passed through a sieve, and stored in sealed bags for analysis. All plants from each sampling point were harvested and categorized into marketable and non-marketable tubers, which were weighed separately to determine the yield.

Soil Sample Collection: The five-point sampling method was used to collect soil from the potato root zone within the plow layer (0–20 cm) under the different treatments. Three replicates from each treatment were thoroughly mixed to form a single composite sample. The collected soil samples were brought back to the laboratory and air-dried naturally. After removing debris, the samples were ground and passed through nylon sieves with mesh sizes of 2.0 mm, 1.0 mm, and 0.149 mm and stored at room temperature for subsequent analysis.

#### 2.4.2. Determination of Soil Physical and Chemical Properties

The soil pH was measured using the potentiometric method with a soil–water ratio of 2.5:1. The electrical conductivity (EC) was determined using 1:5 soil–water extraction and measured with a FE30 Mettler conductivity meter. The soil organic matter (OM) content was assessed using the K_2_Cr_2_O_7_-H_2_SO_4_ digestion method, followed by titration with FeSO_4_. Alkaline nitrogen was measured using the alkaline diffusion method (DB51/T1875-2014). The available phosphorus was determined using the NaHCO_3_ method (NY/T1121.7-2014). The available potassium was extracted with ammonium acetate and measured using flame photometry. The content of Cd and Pb in the soil was determined according to GB/T 17141-1997 [33]: “Determination of lead and cadmium in soil by graphite furnace atomic absorption spectrophotometry”. For the determination of the bioavailable Cd and Pb in the soil, a 5.00 g air-dried soil sample was sieved through a 2 mm mesh and placed in a 100 mL Erlenmeyer flask. Then, 25.00 mL of DTPA extraction solution was added. The flask was placed on a horizontal reciprocal shaker at room temperature (approximately 25 °C) and shaken at 180 strokes per minute for 2 h. After extraction, the mixture was centrifuged or filtered, and 5 mL to 6 mL of the filtrate was collected for analysis. For the determination of the Cd and Pb content in the plants, according to GB5009.268-2016 [34], “Determination of multi-elements in food safety national standard food”, microwave digestion was performed using nitric acid–perchloric acid [35]. Specifically, a 0.5 g sample of plant tissue was placed in a digestion tube. Five milliliters of concentrated nitric acid and perchloric acid was added to the tissue and it was allowed to soak overnight. The digestion tube was then placed in a heating block, where the temperature was maintained at 120 °C for 5 min, followed by 150 °C for 5 min, and finally at 180 °C for 20 min. After cooling the digestion tube, the plant tissue was filtered through a 0.45 μm filter membrane, and deionized water was used to rinse the tube to maintain a constant final volume of 100 mL for the filtrate. The concentrations of cadmium (Cd) and lead (Pb) in the filtrate were determined using an inductively coupled plasma mass spectrometer (ICP-MS), allowing for the measurement of the Cd and Pb content in different parts of the potato. Each batch of samples was digested with 3 blanks and standard samples. Quality control was carried out with the national standard material soil sample (GBW07451) and plant sample GBW10014a (GSB-5a) [28], and the recovery rates of Cd and Pb were 91–110%. The recovery of internal standard element rhodium (Rh) was 91–110%. The test results for the standard samples were within the allowable error range.

#### 2.4.3. Determination of Potato Quality

The determination of the soluble sugar content in the potato was conducted using the anthrone–sulfuric acid method. The determination of the starch content was conducted using acid hydrolysis followed by titration with sodium thiosulfate [36]. For the determination of the vitamin C (VC) content, the 2,6-dichlorophenol indophenol titration method was employed [37]. A 10 g sample was combined with 5 mL of a 2% oxalic acid solution and ground. The mixture was then filtered, and 10 mL of the filtrate was titrated with a standardized 2,6-dichlorophenol indophenol solution until a pink color was achieved. The vitamin C content was calculated based on the volume of the solution used and expressed in mg·100 g^−1^.

### 2.5. Data Analysis

The data obtained from the experiment were averaged over three replicates and summarized using Microsoft Excel 2019. A one-way analysis of variance (ANOVA) was conducted using SPSS 26.0, and graphs were created with Origin 2022. A correlation analysis was performed to examine the relationships between the physicochemical properties of the soil in the potato cultivation area (pH, organic matter (OM), alkaline nitrogen (AN), available phosphorus (AP), available potassium (AK), and bioavailable cadmium (A-Cd) and lead (A-Pb)) and the cadmium (Cd) and lead (Pb) content in different parts of the potato (root, stem, leaf, skin, flesh) using Pearson correlation analysis. The formula for the calculation of the accumulation coefficients of Cd and Pb in potato is as follows:*BCF* = *Cr*/*Cs*(1)

In the equation, *Cr* denotes the heavy metal content in each part of the potato, mg·kg^−1^; *Cs* denotes the heavy metal content in the corresponding inter-root soil of the potato, mg·kg^−1^.

The Cd and Pb transfer coefficients for each part of the potato in this experiment were calculated using the following equations:*TF_Root-Stem_* = *C_Root_*/*C_Stem_*(2)
*TF_Stem-Leaf_* = *C_Leaf_*/*C_Stem_*(3)
*TF_Root-Skin_* = *C_Skin_*/*C_Root_*(4)
*TF_Skin-Flesh_* = *C_Flesh_*/*C_Skin_*(5)

In Equations (2)–(5), TF root-stem, TF stem-leaf, TF root-skin, and TF skin-flesh are the transfer coefficients of Cd or Pb from the root to stem, stem to leaf, root to skin, and skin to flesh in the test potato, respectively; *C_root_*, *C_stem_*, *C_leaf_*, *C_skin_*, and *C_flesh_* denote the content of Cd or Pb in the roots, stems, leaves, skin, and flesh of the potato, respectively, and are given in mg·kg^−1^.

## 3. Results

### 3.1. Physicochemical Properties of Soil

The physicochemical properties of the soil in the potato cultivation area under the different treatments are shown in Table 3. With the exception of treatments T2 and T3, all treatments increased the soil pH. Among these, treatment T6 had the most significant effect in terms of elevating the soil pH, increasing it by 1.75 units compared to the control (CK). Additionally, all treatments significantly improved the organic matter (OM), alkaline nitrogen (AN), available phosphorus (AP), and available potassium (AK) content in the soil compared to CK. Treatment T3 showed the greatest increase in OM, with a 93% enhancement over CK. Treatment T5 was most effective in improving the AN, AP, and AK, with increases of 111%, 276%, and 80%, respectively, compared to CK.

### 3.2. Bioavailable Cadmium (Cd) and Lead (Pb) Content in Potato Rhizosphere Soil

The combined application of different passivators and their single applications had different effects on the content of available Cd and Pb in the potato soil, as shown in Figure 1. Compared with CK, the combined application and single application of different passivators could reduce the content of available Cd and Pb. Compared with the T4 and T5 treatments, T6 had the best passivation effect regarding Cd and Pb; it significantly reduced the soil available Cd content by 1.22% and 4.71% and the Pb content by 3.13% and 3.02%, respectively. Biochar usually has a large specific surface area and abundant pores, which enable it to effectively adsorb heavy metal ions [31]. The calcium ions contained in calcium magnesium phosphate (Ca^2+^) can complex with the heavy metal ions in soil to form insoluble compounds or precipitates [30]. The synergistic effect of calcium magnesium phosphate fertilizers and biochar results in the better mitigation of heavy metals. The carrier effect of biochar prolongs the release time of calcium magnesium phosphate, rendering the passivation effect on the heavy metals in the soil more long-lasting, thus reducing their bioavailability.

### 3.3. Effects of Different Treatments on Cd and Pb Content in Various Parts of Potato

The different treatments had certain effects on the Cd content in different parts of potato, as shown in Table 4. In all treatments, compared with CK, the T4 treatment showed the most significant difference in reducing the potato root content, which was significantly lower than CK by 60.76%. Compared with T5, the T6 treatment showed the most significant difference in reducing the Cd content in the potato stems, leaves, peels, and flesh, which was significantly reduced by 23.17%, 17.10%, 56.08%, and 31.02%, respectively. Compared with CK, all treatments significantly reduced the Pb content in all parts of potato. Compared with T5, the T6 treatment showed the most significant difference in reducing the Pb content in the potato roots, stems, peels, and flesh, which was significantly lower, namely 0.83%, 6.62%, 6.81%, and 20.00%, respectively. The content of Cd and Pb in potato flesh, ranging from high to low, was in the order of CK > T2 > T1 > T4 > T3 > T5 > T6. It can be seen that the T6 treatment had the best effect in terms of reducing the content of Cd and Pb in each part of the potato, followed by the T5 treatment. The Cd content in the potato pulp and the Pb content in the stem, leaf, skin, and edible parts of all treated potatoes reached the maximum limit values of the Chinese feed pollutant limit standard GB 13078-2017 [38] (Cd ≤ 1 mg·kg^−1^, Pb ≤ 30 mg·kg^−1^).

### 3.4. Effects of Different Treatments on Cd and Pb Accumulation Coefficients in Various Parts of Potato

The bioaccumulation coefficient (BCF) refers to the ratio of the average concentration of a specific element in different plant tissue types to its average concentration in the soil. In particular, the BCF indicates the ability of potatoes to absorb and accumulate cadmium (Cd) and lead (Pb) from the soil; a higher BCF reflects a greater capacity for the accumulation of these heavy metals [39]. Compared with CK, the different treatments had a certain effect on the capacity for the enrichment of Cd and Pb in different parts of potato, which can be seen in Figure 2. The T4 treatment showed the most significant difference in reducing the BCF of Cd in the potato roots. The T5 treatment showed the most significant difference in reducing the BCF of Pb in the potato roots and leaves. Compared with T4 and T5, the T6 treatment showed the most significant difference in reducing the BCF of Cd in the potato stems, leaves, peels, and flesh, and it reduced the BCF of the potato flesh by 50% and 30.00%, respectively. Compared with T4 and T5, the T6 treatment showed the most significant difference in reducing the Pb enrichment coefficient of the potato stem, peel, and potato flesh; it reduced the BCF of the potato flesh by 18.75% and 7.14%, respectively. The enrichment capacity of Cd and Pb in each part from large to small was in the order of root > stem > leaf > peel > flesh. It can be seen that the T6 treatment had the best effect in terms of reducing the BCF of Cd and Pb in various parts of potato, followed by the T5 treatment.

### 3.5. Effects of Different Treatments on Transport Coefficients of Cd and Pb in Various Parts of Potato

The transfer coefficient (TF) refers to the ratio of the average content of a certain element in the above-ground part of the plant to its average content in the below-ground part; a higher transfer coefficient indicates a stronger ability for heavy metals to migrate to the next organ. Compared with CK, the different treatments had different effects on the translocation coefficients of Cd and Pb in different parts of potato, as shown in Figure 3. The T6 treatment showed the most significant difference in reducing the root–stem TF and root–skin TF of potato for Cd; they were significantly reduced by 28.42% and 51.16%, respectively, compared with CK. The results also showed that the T6 treatment effectively inhibited this type of transfer in potato from root to stem and peel, while the other treatments showed no significant differences regarding the peel–flesh TF for Cd. The T4 treatment reduced the stem–leaf TF of potato for Cd to the greatest extent; it was 19.04% lower than that of CK, indicating that the T4 treatment effectively inhibited this type of transfer from stem to leaf. Compared with CK, all passivator treatments reduced the TF of Pb from root to stem, and the T6 treatment showed the most significant difference in reducing the root–stem TF, root–skin TF, and skin–flesh TF of potato for Pd compared with T4. They were significantly reduced by 7.65%, 5.30%, and 10.70%, respectively. Moreover, compared with T5, they were significantly reduced by 5.83%, 6.02%, and 7.38%, respectively. This indicates that the application of the T6 treatment effectively inhibited the root–stem, root–skin, and skin–flesh transfer of Pd in potato, while the remaining treatments showed no significant difference regarding the skin–flesh TF for Pd in potato.

### 3.6. Correlation Analysis

The Pearson correlation analysis of the physicochemical properties of the soil in the potato planting area (pH, OM, AN, AP, AK, A-Cd, and A-Pb) with the Cd and Pb content in different parts of the potato (root, stem, leaf, skin, flesh) indicated that the soil AP was significantly and negatively correlated with the Cd content in the potato leaves and flesh (*p* < 0.05). Additionally, the soil’s effective cadmium content (A-Cd) showed a significant and positive correlation with the Cd content in the potato roots (*p* < 0.05) (Figure 4). Moreover, the soil AP exhibited a highly significant and negative correlation with the Pb content in the potato roots and leaves (*p* < 0.01), as well as a significant and negative correlation with the Pb content in the stems, skin, and flesh (*p* < 0.05). The soil’s effective lead content (A-Pb) was significantly and positively correlated with the Pb content in the potato stems and flesh (*p* < 0.05) and showed a highly significant and positive correlation with the root Pb content (*p* < 0.01) (Figure 5). In summary, the effective lead and cadmium content in the soil is a key factor in the absorption and accumulation of lead and cadmium in various parts of the potato. Additionally, the soil effective phosphorus has an antagonistic, competitive relationship with the effective cadmium content. The effective phosphorus content in the soil can influence the effective cadmium and lead levels, thereby determining the absorption and accumulation of lead and cadmium in different parts of the potato.

### 3.7. Effects of Different Treatments on Potato Yield and Quality

The different treatments had varying effects on the quality and yield of the potatoes, as shown in Table 5. Compared to the control (CK), all treatments significantly increased the potato yield, with the T1 treatment achieving the highest yield of 28,716.64 kg·hm^−2^. Among all treatments, T1 had the best effect in increasing the vitamin content, raising it significantly by 33.90% compared to CK, while the T5 treatment was the most effective in increasing the starch content, with a significant increase of 12.69% compared to CK.

## 4. Discussion

This study found that the application of various passivators reduced the content of available Cd and Pb in soil (Figure 1) and the content of Cd and Pb in various parts of potato (Table 4) to varying degrees. Firstly, this is because the application of different amendments to the soil significantly increased its pH. An increase in the soil’s pH value is beneficial to increase the negative charge of the soil and increase the adsorption, complexation, and even precipitation of Cd and Pb in the soil solution by soil particles, thus effectively reducing the availability of Cd and Pb in the soil [40] and reducing the accumulation of heavy metals in crops [41,42,43]. Secondly, all passivator treatments could increase the soil nutrient content (Table 3), among which the single application of the calcium magnesium phosphate fertilizer (T5) yielded the best improvement in the soil available nutrients. The calcium magnesium phosphate fertilizer is an alkaline citrate–soluble phosphate fertilizer with a gentle fertilizing effect and high phosphorus utilization rate. At the same time, it can increase the soil pH and the content of available phosphorus, calcium, magnesium, and other nutrients, which are generally lacking in southern soils. The research by Duan et al. [44] showed that a calcium magnesium phosphate fertilizer could improve the utilization rate of phosphorus in the rhizosphere soil of rhubarb and improve the physical and chemical properties of the soil. Therefore, it is often used as a passivator to reduce the content of heavy metals in soil [45]. Yang et al. [46] demonstrated that the application of a calcium magnesium phosphate fertilizer could increase the soil pH and reduce the bioavailability of heavy metals. Additionally, the calcium magnesium phosphate fertilizer reacts with cadmium (Cd) through its abundant phosphate ions to form insoluble phosphate precipitates, which inhibit the migration and transformation of Cd within the soil system [47]. The calcium magnesium phosphate fertilizer can weaken the effect of exchangeable Al^3+^ and H^+^ on soil acidification by supplementing the exchangeable Ca^2+^ and Mg^2+^ in the soil; it can also reduce the content of available and acid-extractable Cd in the soil, increase the content of carbonate-bound and iron–manganese oxide-bound Cd, reduce the effectiveness of Cd, and increase the adsorption and fixation of Cd in the soil [48].

In many cases, the remediation effect of a single passivator is poor, particularly when dealing with more complex or heavily contaminated soil. For example, the calcium magnesium phosphate fertilizer is rich in phosphorus, calcium, magnesium, silicon, and other elements, and its excessive application may cause soil nutrient imbalances. With the passage of time, the pore structure of biochar may be filled with fine particles or organic matter, resulting in a decrease in the specific surface area and adsorption capacity. The combined application of different passivators can exploit the advantages of the chemical passivation of inorganic materials, pH regulation, and organic materials to improve the absorption and transport of pollutants by plants and promote crop growth [49]. In this study, the combined application of biochar and calcium magnesium phosphate fertilizer (T6) had the best remediation effect on Cd–Pb-contaminated soil, because the pH regulation ability of the composite material was the best, and the pH is the key factor affecting the passivation of heavy metals in soil. Precipitation/co-precipitation and adsorption are the main mechanisms, but ion exchange, electrostatic interaction, surface complexation, and other mechanisms are also combined in the soil–composite interaction system [47]. Jun et al. [50] showed that the addition of biochar and a calcium magnesium phosphate fertilizer to heavy metal-contaminated soil could significantly increase the pH value and enhance the heavy metal remediation effect in sunflower; this is similar to the results of the present study. The different properties of different passivators lead to different effects in terms of reducing the bioavailability of Cd and Pb in the soil. Yuan et al. [51] studied the remediation effects of different amounts of sepiolite, lime, humic acid, biochar, and calcium magnesium phosphate fertilizer on corn farmland around a mining area through pot and field experiments. The results showed that sepiolite and lime had the best remediation effects on Cd and Pb. This shows that the effects of biochar and calcium magnesium phosphate treatment on potato are significantly different from those on corn. Wu et al. [52] discovered that the application of an inorganic phosphorus fertilizer could reduce the heavy metal concentrations in maize plants. This study also confirmed that there was a significant and negative correlation between the soil bioavailable phosphorus and the cadmium and lead levels in potato. Conversely, the soil bioavailable Cd and Pb levels exhibited a significant and positive correlation with the cadmium and lead concentrations in potato. Furthermore, there is an antagonistic, competitive relationship between the soil available phosphorus and the available cadmium and lead concentrations in the soil (Figure 4 and Figure 5). Additionally, research has indicated that, during the maturation phase, the Cd and Pb accumulation capacity in different potato parts is ranked from highest to lowest as follows: roots > stems > leaves > skins > flesh. This suggests that the potato roots have the highest concentrations of Cd and Pb, making them the most susceptible to heavy metal accumulation. This phenomenon may be attributed to the important mechanisms of heavy metal ion transport across vacuolar membranes and their fixation within vacuoles and various plant tissue types [53,54]. Potato has a strong ability to absorb, transport, and enrich Cd and Pb in soil. The content of Cd in potato pulp and Pb in the stem, leaf, skin, and edible parts of potato when treated with all passivators met the maximum limit values specified in the Chinese feed pollutant limit standard GB 13078-2017 [38] (Cd ≤ 1 mg·kg^−1^, Pb ≤ 30 mg·kg^−1^). However, the content of Cd and Pb in the potato flesh exceeded the recommendation of the food limit standard GB2763-2022 [28] (Cd ≤ 0.1 mg·kg^−1^, Pb ≤ 0.2 mg·kg^−1^). The strong absorption and enrichment ability of potato regarding Cd and Pb may be related to the high total amounts of Cd and Pb in the soil, in addition to the variety.

Humic acid is derived from natural organic matter and can enhance soil fertility. Raw biochar materials are derived from the high-temperature pyrolysis of agricultural and forestry biomass waste. This not only effectively passivates soil heavy metals and reduces crop absorption but is also conducive to soil carbon sequestration and emission reduction, improving the soil’s physical and chemical properties and improving crop yields and quality. It has good agricultural and environmental effects [55]. In this study, humic acid, biochar, and calcium magnesium phosphate fertilizer were used to explore the effects of the combined application of different passivators and their single application on the accumulation of lead and cadmium in potato, as well as the yield and quality of potato. Among them, biochar combined with calcium magnesium phosphate fertilizer (T6) had a certain effect on the remediation of heavy metal pollution in farmland around the Yunnan mining area in China, as well as enabled improvements in yield and quality. However, this study lacked seasonal data and failed to consider the potential impact of the long-term combined application of biochar and calcium magnesium phosphate (T6) on the microbial community. Moreover, the effective state and regulation mechanisms of Cd and Pb in potato or other crop soils around other mining areas still need further study.

## 5. Conclusions

This study shows that the combined application of different amendments can effectively improve the physical and chemical properties of soil in contaminated mining areas and significantly reduce the content of available Cd and Pb in soil and their accumulation in various parts of potato. Among them, the combined application of biochar and the calcium magnesium phosphate fertilizer was superior to the individual application method in reducing the content of available Cd and Pb in the soil and the content of Cd and Pb in the potato flesh. The synergistic effect of the biochar and calcium magnesium phosphate fertilizer not only increased the content of available nutrients in the soil, but also effectively inhibited the activity of heavy metals, thus reducing the transfer and accumulation of Cd and Pb in potato and improving the yield and quality of the crop. This study provides important technical support for the safe planting of potatoes and agricultural soil management, and it also provides a reference solution for the planting of other crops in heavy metal-contaminated soil, especially in those with similar environmental backgrounds. Future research should focus on different soil types, climatic conditions, and crop varieties and further test the effects of various combinations of amendments, so as to provide more comprehensive research ideas and practical guidance to address the problem of heavy metal pollution in agricultural soils.

## Figures and Tables

**Figure 1 life-14-01615-f001:**
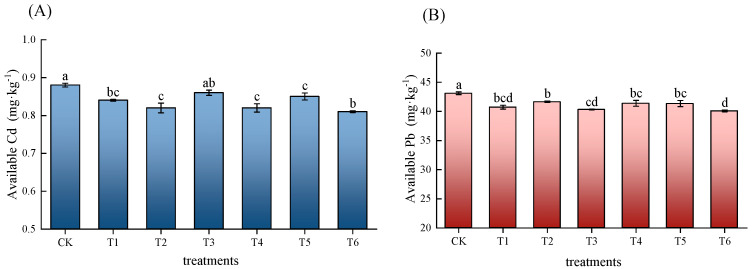
Effects of different treatments on the effective state Cd and Pb content of potato inter-root soil. (**A**) Effective available Cd content of potato inter-root soil; (**B**) effective available Pb content of potato inter-root soil. Each column indicates the mean of three replicates (*n* = 3) and the error line indicates the standard deviation. Different letters indicate significant differences (*p* < 0.05).

**Figure 2 life-14-01615-f002:**
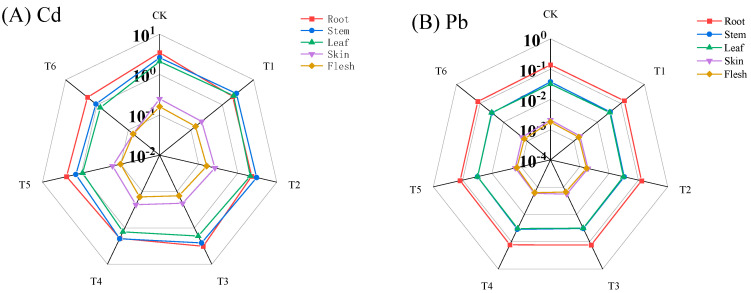
Effects of different fertilization treatments on the enrichment coefficients of Cd and Pb in different parts of potato. (**A**) Cd enrichment coefficient of each part of potato; (**B**) Pb enrichment coefficient of each part of potato. Each number is the average of 3 replicates (n = 3), *p* < 0.05.

**Figure 3 life-14-01615-f003:**
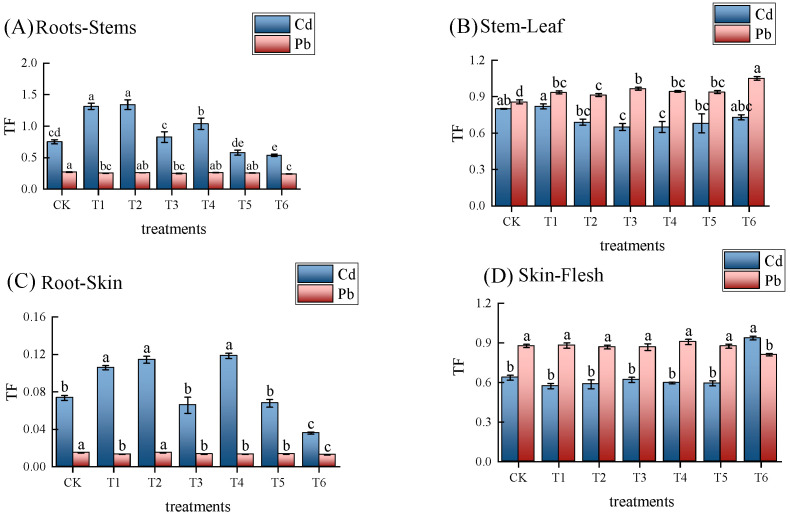
Illustrates the effects of different fertilization treatments on the transfer coefficients of Cd and Pb in various parts of potato. TF: transfer coefficient. (**A**) indicates the transfer coefficient from the root to the stem in potato; (**B**) indicates the transfer coefficient from the stem to the leaf; (**C**) indicates the transfer coefficient from the root to the skin; (**D**) indicates the transfer coefficient from the skin to the flesh. Each column represents the mean value of three replicates (*n* = 3), and the error bars indicate the standard deviation. Different letters denote significant differences (*p* < 0.05).

**Figure 4 life-14-01615-f004:**
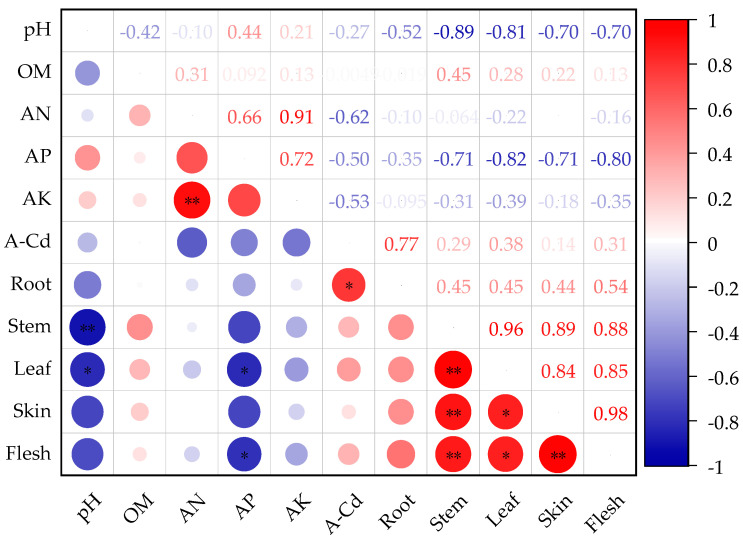
Correlation analysis between Cd content in different parts of the potato and soil physicochemical properties. * indicates *p* < 0.05; ** indicates *p* < 0.01.

**Figure 5 life-14-01615-f005:**
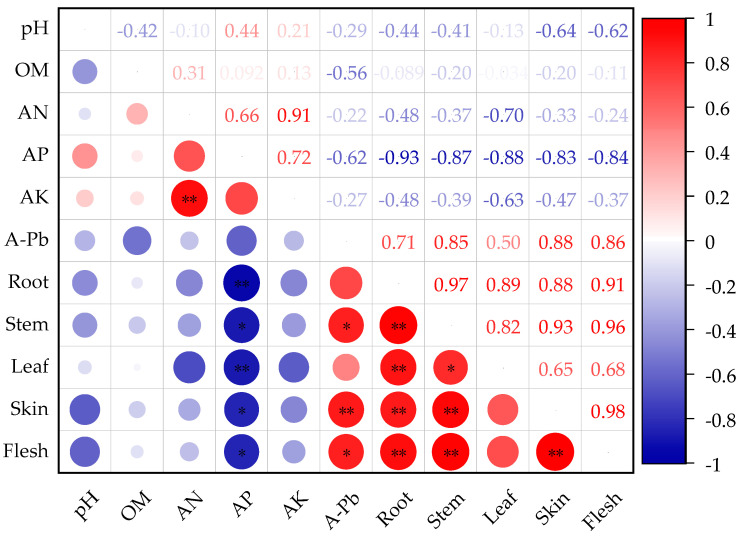
Correlation analysis between Pb content in different parts of the potato and soil physicochemical properties. * indicates *p* < 0.05, ** indicates *p* < 0.01.

**Table 1 life-14-01615-t001:** Basic physical and chemical properties of tested soil.

pH	CEC	OM	AN	AP	AK	Cd	Pb	Soil Texture (%)		
	(cmo·kg^−1^)	(g·kg^−1^)	(mg·kg^−1^)	(mg·kg^−1^)	(mg·kg^−1^)	(mg·kg^−1^)	(mg·kg^−1^)	Sand	Clay	Silt
5.36	18	19.99	153.73	15.64	393	6.42	273.69	20	40	30

**Table 2 life-14-01615-t002:** Field fertilization treatments.

Treatments	Treatment Establishment	Usage and Dosage
CK	Control	---
T1	Special potato fertilizer (16-10-6)	1500 kg·hm^−2^
T2	Humic acid (>20%)	1500 kg·hm^−2^
T3	Potato fertilizer + humic acid	1500 kg·hm^−2^ (special fertilizer–soil conditioner 1:1)
T4	Biochar (15-7-18)	1500 kg·hm^−2^
T5	Calcium magnesium phosphate (P_2_O_5_ ≥ 12%)	1500 kg·hm^−2^
T6	Biochar + calcium magnesium phosphate	1500 kg·hm^−2^ (Biochar–calcium magnesium phosphate 1:1)

**Table 3 life-14-01615-t003:** Physicochemical properties of soil.

Treatments	pH	OM (g·kg^−1^)	AN (mg·kg^−1^)	AP (mg·kg^−1^)	Ak (mg·kg^−1^)
CK	6.12 ± 0.02 e	18.39 ± 0.80 e	101.34 ± 13.07 d	14.26 ± 1.02 f	191.33 ± 6.11 e
T1	6.34 ± 0.04 d	33.44 ± 0.90 b	139.92 ± 16.16 c	22.82 ± 2.12 e	241 ± 5.29 c
T2	5.2 ± 0.04 b	31.67 ± 0.51 bc	161.89 ± 16.63 c	30.59 ± 1.71 d	214 ± 7.00 d
T3	5.8 ± 0.03 a	35.54 ± 1.21 a	184.54 ± 9.33 b	43.3 ± 0.93 b	281.33 ± 3.21 b
T4	7.76 ± 0.02 c	30.73 ± 2.36 c	147.31 ± 11.40 c	33.2 ± 0.95 c	245.67 ± 12.01 c
T5	7.42 ± 0.03 f	20.2 ± 1.04 e	214.17 ± 9.06 a	53.64 ± 1.13 a	342.67 ± 9.71 a
T6	7.87 ± 0.03 e	24.12 ± 1.01 d	105.38 ± 8.95 d	43.65 ± 1.58 b	202.33 ± 6.66 de

Note: OM, organic matter; AN, alkaline nitrogen decomposition; AP, effective phosphorus; AK, effective potassium. Data are the means of three replicates (means ± standard errors). Different lowercase letters indicate significant differences between the treatments at *p* < 0.05.

**Table 4 life-14-01615-t004:** The contents of Cd and Pb in different parts of potato under different treatments.

	Treatment	Root (mg·kg^−1^)	Stem (mg·kg^−1^)	Leaf (mg·kg^−1^)	Skin (mg·kg^−1^)	Flesh (mg·kg^−1^)
	CK	15.37 ± 1.23 a	11.52 ± 0.04 a	9.23 ± 0.08 b	1.13 ± 0.14 a	0.72 ± 0.11 a
	T1	9.17 ± 0.12 c	12.04 ± 0.66 a	9.86 ± 0.29 a	0.97 ± 0.02 ab	0.54 ± 0.01 b
	T2	9.34 ± 0.31 c	12.49 ± 0.85 a	8.62 ± 0.11 c	1.06 ± 0.02 ab	0.57 ± 0.02 b
Cd	T3	13.65 ± 1.33 b	11.12 ± 0.89 a	7.20 ± 0.30 d	0.88 ± 0.19 b	0.47 ± 0.11 b
	T4	7.95 ± 0.30 c	8.20 ± 0.94 b	5.27 ± 0.33 e	0.94 ± 0.03 ab	0.50 ± 0.02 b
	T5	9.81 ± 0.48 c	5.68 ± 0.37 c	3.81 ± 0.54 f	0.66 ± 0.01 c	0.35 ± 0.04 c
	T6	8.37 ± 0.43 c	4.50 ± 0.14 d	3.26 ± 0.06 g	0.30 ± 0.02 d	0.25 ± 0.01 d
	CK	39.99 ± 0.25 a	10.79 ± 0.46 a	9.23 ± 0.11 a	0.53 ± 0.01 a	0.34 ± 0.06 a
	T1	37.49 ± 0.41 b	9.52 ± 0.075 b	8.90 ± 0.21 abc	0.49 ± 0.02 c	0.28 ± 0.03 bc
	T2	35.57 ± 0.38 c	9.27 ± 0.077 b	8.47 ± 0.12 cd	0.50 ± 0.01 b	0.29 ± 0.01 b
Pb	T3	35.44 ± 0.76 c	8.88 ± 0.095 c	8.57 ± 0.11 bcd	0.47 ± 0.01 d	0.26 ± 0.02 cd
	T4	36.22 ± 0.87 c	9.51 ± 0.13 b	8.97 ± 0.13 ab	0.47 ± 0.04 d	0.29 ± 0.04 bc
	T5	33.61 ± 0.56 d	8.653 ± 0.15 c	8.12 ± 0.04 d	0.44 ± 0.06 e	0.25 ± 0.06 d
	T6	33.33 ± 0.84 d	8.08 ± 0.18 d	8.50 ± 0.41 cd	0.41 ± 0.01 f	0.20 ± 0.01 e

Note: Mean ± standard deviation; different lowercase letters in the same column indicate significant (*p* < 0.05) differences in Cd and Pb content of potato parts between different fertilization treatments.

**Table 5 life-14-01615-t005:** Effects of different treatments on yield and quality of potato.

Treatment	Yield (kg·hm^−2^)	Vitamin Content (mg·100 g^−1^)	Starch Content (g·100 g^−1^)	Contents of Soluble Sugar (g·100 g^−1^)
CK	21,090.67 ± 1530.04 c	10.08 ± 0.06 c	15.53 ± 0.42 d	2.50 ± 0.04 a
T1	28,716.64 ± 1005.94 a	13.50 ± 0.34 a	14.90 ± 0.32 e	1.60 ± 0.01 c
T2	22,850.64 ± 1477.19 bc	10.26 ± 0.23 c	15.79 ± 0.60 c	2.26 ± 0.09 b
T3	23,621.61 ± 809.76 abc	10.43 ± 0.03 c	17.12 ± 1.14 b	2.70 ± 0.14 a
T4	27,795.48 ± 849.16 ab	12.00 ± 0.04 b	15.61 ± 0.55 cd	2.48 ± 0.07 a
T5	26,479.52 ± 568.90 ab	11.55 ± 0.05 b	17.50 ± 0.70 a	2.52 ± 0.05 a
T6	24,312.83 ± 1022.61 abc	10.41 ± 0.34 c	14.75 ± 0.23 e	2.53 ± 0.18 a

Note: Mean ± standard deviation; different lowercase letters within the same column indicate significant differences in potato yield and quality among different fertilization treatments (*p* < 0.05).

## Data Availability

Data are contained within the article.

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
