# Peer review of "Reducing Cd and Pb Accumulation in Potatoes: The Role of Soil Passivators in Contaminated Mining Soils"

_life, 2024, doi:10.3390/life14121615_

Round 1
Reviewer 1 Report
Comments and Suggestions for Authors
1. Double-check for spelling and spacing errors throughout manuscript to improve readability.
2. Define all abbreviations at their first mention and use consistently thereafter to maintain readability and avoid redundancy.
3. Emphasize the novelty of this study by comparing the combined treatments impact on Cd-Pb remediation with single-passivators.
4. Discuss whether reduced Cd and Pb levels meet safety standards, considering potential health implications for human consumption.
5. References need minor formatting adjustments.
6. Abstract: Rephrasing the aim to be more concise with the impact of T6 on yield and quality alongside soil improvements for a balanced summary.
7. Introduction: Include a few words on why potato is especially suitable for such studies beyond its economic value.
8. Materials and Methods: Add biochar details whether it is commercially obtained or produced in the lab (resource, production temperature, and properties).
9. Potato seeds or seedlings (days?) were used in field trials to ensure clear experimental setup and reproducibility.
10. Discussion: Add few lines on the environmental sustainability of using these passivators long-term in mining areas.
11. Explain selection criteria for each passivating agent, especially their specific benefits for Cd and Pb immobilization in soils.
12. Remove lines 465-468.
13. Rewrite lines 470-471 for Clarity
14. How biochar with calcium-magnesium phosphate synergy enhances Cd-Pb immobilization for better soil remediation results?
15. Format table 6.
Reviewer 2 Report
Comments and Suggestions for Authors
Review of the Manuscript: "Study on the Inhibition of Cd and Pb Accumulation in Potato by Different Passivators in Remediation of Cd and Pb Contaminated Soil in Mining Area"
1. Title and Abstract
Title: The title accurately reflects the study's objective, yet it could be refined for conciseness. A suggestion could be: “Reducing Cd and Pb Accumulation in Potatoes: The Role of Soil Passivators in Contaminated Mining Soils.”
Abstract:
Structure and Relevance: The abstract provides a comprehensive overview; however, it introduces some minor redundancies that detract from clarity. In a few sentences, combining results with broader implications could enhance readability.
Results: The level of detail in the results section is excessive for an abstract. A tighter focus on only the primary findings—such as the standout effect of treatment T6—would improve brevity and impact.
Conclusion: While the conclusion is informative, a more generalized statement on the applicability of these findings to food safety or soil management in agriculture would expand the relevance.
2. Introduction
Context: The introduction effectively explains the issue of soil contamination due to mining activities. However, it would be beneficial to briefly discuss contamination impacts beyond China, lending the study broader global importance.
Background Literature: There is a good foundation, but the section could emphasize current gaps in knowledge, particularly regarding the effectiveness of combined passivator applications. Highlighting what is unknown about different passivator combinations would strengthen the need for this study.
Objectives: The objectives are somewhat implied but not distinctly stated. Reframing them as clear, direct research questions would enhance focus and guide readers on what this study aims to achieve.
3. Materials and Methods
Study Area Details: While the area is well-characterized in terms of climate and soil type, additional background data on baseline heavy metal concentrations, soil texture, and cation exchange capacity would improve context and reproducibility.
Experimental Design:
Method Justification: A randomized block design is appropriate; however, justifying plot sizes, replication, and especially the selection of potato variety could clarify the design choices. These details are crucial for establishing methodological rigor.
Description of Passivators: Each passivator (biochar, humic acid, etc.) should be described in terms of chemical composition, source, and functional properties, such as pH effect or specific heavy metal binding capacity. These characteristics directly impact their remediation effectiveness.
Analytical Techniques:
Soil and Plant Analyses: While the analytical methods are generally sound, including references to standard protocols (such as USEPA or ISO guidelines) would boost credibility. Details on calibration and precision (e.g., blank samples, recovery rates) should be provided for ICP-MS measurements.
Quality Assurance: Methodological accuracy would benefit from additional details on quality control measures for heavy metal analyses in both soils and plant tissues, specifically in terms of instrumental calibration and sample handling.
4. Results
Data Presentation:
Tables and Figures: The tables effectively communicate treatment differences, but greater use of visual representations (e.g., line plots or box plots) could make comparisons between treatments more intuitive. Simplifying complex tables where possible can also aid clarity.
Statistical Reporting: The use of ANOVA is appropriate, but post-hoc test results to clarify specific treatment differences are missing. Including effect sizes would strengthen interpretation, particularly in relation to Cd and Pb reduction.
Detailed Treatment Impact: Results indicate that treatment T6 (biochar + calcium-magnesium phosphate) showed superior effects in reducing metal bioavailability. However, additional analysis on how biochar’s structure (e.g., pore size) or calcium’s role contributes to this outcome would enhance the reader’s understanding of why these materials outperform others.
Interpretation:
Effectiveness of Passivators: The results highlight differences in treatment efficacy, but there is little discussion on the underlying reasons for these differences. Providing more insight into the chemical interactions of Cd and Pb with biochar and calcium-magnesium phosphate would make this clearer.
Comparative Context: A comparison with similar studies on other crops or regions would add a broader perspective to the findings, especially if treatments show unique effectiveness for potatoes compared to other crops.
5. Discussion
Depth of Analysis:
The discussion is thorough but could go further in analyzing the specific limitations and challenges of using passivators. Discussing the stability of biochar over time and the risk of nutrient imbalances from high phosphates could be useful, as these are practical issues in field application.
Mechanistic Insights: The discussion would benefit from a closer examination of the molecular and biochemical mechanisms that reduce heavy metal availability, particularly interactions with soil pH and organic content.
Limitations: This section would feel more robust with explicit acknowledgment of study limitations, such as the lack of seasonal data or the potential impact of treatments on microbial communities. Recognizing these could lend a more balanced view.
6. Conclusion
Clarity and Broader Implications: The conclusion is concise but could be improved by emphasizing the practical applications for soil and crop management, particularly in relation to food safety and remediation practices. Adding a sentence on potential applicability to other crops or environmental contexts would broaden the impact.
Future Directions: The authors should specify areas for future research, such as testing the passivator combination in different soil types, climatic conditions, or crop varieties. This would suggest a comprehensive approach to addressing the heavy metal contamination issue in agricultural soils.
7. References
Citations and Currency: The manuscript could benefit from including more recent studies, particularly regarding the latest advancements in passivator technology and soil remediation. Adding recent, peer-reviewed references from the last five years would provide an up-to-date foundation for the research.
Accuracy: Some inconsistencies in in-text citations suggest the need for careful editing to ensure all citations are appropriately matched with statements in the text.
Overall Summary
This manuscript provides valuable insights into using passivators for reducing Cd and Pb accumulation in potatoes grown in contaminated soils. However, the technical rigor could be enhanced by including mechanistic explanations, citing recent literature, and discussing practical field challenges in greater detail.
Comments on the Quality of English LanguageEnglish editing is required throughout manuscript.
Reviewer 3 Report
Comments and Suggestions for Authors
Manuscript Title: Study on the inhibition of Cd and Pb accumulation in potato by different passivators in remediation of Cd and Pb contaminated soil in mining area
Manuscript ID: life-3311725
Comments: Major revision.
Title: Change it to a more meaningful title.
Abstract:
1. “To explore safe utilization techniques for farmland surrounding mining areas contaminated by heavy metals, specifically cadmium (Cd) and lead (Pb), in order to achieve food security.”. The sentence is not correct grammatically.
2. “A field experiment was conducted using potatoes as the test crop, with seven treatments established”: -not grammatically correct.
3. Please check throughout the manuscript to improve English.
4. Line 36: “In summary”.-abstract itself the summary of the study. Remove this type of words from the abstract.
Introduction:
1. The Introduction section needs to be modified by describing the cause, fatality, novelty of the work, advantages, and disadvantages of the methods applied. There are many studies reported in the literature regarding heavy metals remediation from soil using diverse materials. The manuscript would be strengthened by clearly defining and testing specific hypotheses. While the study provides extensive data and analysis, framing the research around a central hypothesis or research question could enhance the coherence and focus of the work. This approach would help guide the experimental design and analysis, providing a more structured narrative.
2. Clarify the Novelty of the work.
Methodology
1. This part is okay however check English during revision.
Results and discussion
1. T6 treatment seems provided the best results. However, the result and discussion section must explain more experimental findings, the significance of work, and mechanisms of heavy metal remediation in soil.
Conclusion
1. This section must be re-written and it should be specific to the results.
Comments on the Quality of English Language
English improvement is required.
Reviewer 4 Report
Comments and Suggestions for Authors
The paper describes the research well with minor English errors. On page 5 row 184, it is unclear what (3.11) is referring to. The paper described different passivator treatments for potato yield and quality. All treatments reduce available Cd and Pb. The synergistic effect of biochar and calcium-magnesium-phosphate fertilizer on the remediation of Cd-Pb composite soil is demonstrated. In addition, the inhibition of Cd and Pb accumulation and transfer within the potato was shown to be effective.
Comments on the Quality of English LanguageThere are issues with the English. Table 2 CK is not contrast but control.
Round 2
Reviewer 3 Report
Comments and Suggestions for Authors
The writing in the manuscript has been improved and the manuscript is accepted in its current form.